# Can smartphone technology be used to support an effective home exercise intervention to prevent falls amongst community dwelling older adults?: the TOGETHER feasibility RCT study protocol

Helen Hawley-Hague,[1] Carlo Tacconi,[2,3] Sabato Mellone,[2,3,4] Ellen Martinez,[5] Angela Easdon,[6] Fan Bella Yang,[7] Ting-Li Su,[8] A Stefanie Mikolaizak,[9] Lorenzo Chiari,[2,3,4] Jorunn L Helbostad,[10] Chris Todd[1]

For numbered affiliations see end of article.

**Correspondence to**
Dr Helen Hawley-Hague; helen.hawley-hague@manchester.ac.uk

## ABSTRACT

**Introduction** Falls have major implications for quality of life, independence and cost to the health service. Strength and balance training has been found to be effective in reducing the rate/risk of falls, as long as there is adequate fidelity to the evidence-based programme. Health services are often unable to deliver the evidence-based dose of exercise and older adults do not always sufficiently adhere to their programme to gain full outcomes. Smartphone technology based on behaviour-change theory has been used to support healthy lifestyles, but not falls prevention exercise. This feasibility trial will explore whether smartphone technology can support patients to better adhere to an evidence-based rehabilitation programme and test study procedures/outcome measures.

**Methods and analysis** A two-arm, pragmatic feasibility randomised controlled trial will be conducted with health services in Manchester, UK. Seventy-two patients aged 50+years eligible for a falls rehabilitation exercise programme from two community services will receive: (1) standard service with a smartphone for outcome measurement only or (2) standard service plus a smartphone including the motivational smartphone app. The primary outcome is feasibility of the intervention, study design and procedures. The secondary outcome is to compare standard outcome measures for falls, function and adherence to instrumented versions collected using smartphone. Outcome measures collected include balance, function, falls, strength, fear of falling, health-related quality of life, resource use and adherence. Outcomes are measured at baseline, 3 and 6-month post-randomisation. Interviews/focus groups with health professionals and participants further explore feasibility of the technology and trial procedures. Primarily analyses will be descriptive.

**Ethics and dissemination** The study protocol is approved by North West Greater Manchester East Research Ethics Committee (Rec ref:18/NW/0457, 9/07/2018). User groups and patient representatives were consulted to inform trial design, and are involved in study recruitment. Results will be reported at conferences and in peer-reviewed publications. A dissemination event will be held in Manchester to present the results of the trial. The protocol adheres to the recommended Standard Protocol Items: Recommendations for Interventional Trials (SPIRIT) checklist.

**Trial registration number** ISRCTN12830220; Pre-results.

## INTRODUCTION

Falls are an important public health issue, with over 30% of people aged 65 and over falling at least once a year.[1] This has implications for quality of life, independence and cost to the health service.[1] Strength and balance training (SBT) comprises 'carrying out exercises that increase muscle strength in the legs and improve balance'.[2] Strength and balance exercise programmes are effective in reducing risk and rate of falls and injuries.[3] Sherrington et al[4] have shown that for strength and balance programmes to be effective they need to be progressive, tailored and of adequate dose (3× a week for 50 hours, and then maintained). Work carried out by Public Health England[5] illustrates that to see a return on investment, fidelity to the evidence-base has to be carried out (adequate dose, progression).

However, Nyman and Victor[6] report that adherence to evidence-based strength and balance programmes is poor. The National Health Service (NHS) only delivers programmes that are predominantly 3 months or less,[7] older adults do not carry out their exercise programme three times a week as prescribed (dose) or carry out the programme for a sufficient length of time to achieve and maintain the benefits.[6,7] Cost and

appropriate staffing are cited as primary reasons for short NHS delivery.[7]

Unless there are innovative new solutions to support the delivery of falls prevention exercises sufficiently to reduce falls risk and to prevent re-referral to services, over the coming decade it is estimated that population changes will result in service demand beyond the reach of current interventions.[8] The use of smartphones to support falls rehabilitation could be one of the solutions. The proportion of older adults using smartphones is growing rapidly, with 39% of those aged 65 to 74% and 15% of those aged over 75 using smartphones.[9] Smartphones offer multiple opportunities to support healthy ageing and falls prevention as they are portable, can be body-worn and can therefore be used for falls detection, movement detection and motivation[10–12] The evidence which looks at the role of the smartphone for falls prevention is sparse,[13] particularly for interventions focused on rehabilitation/SBT. Although there is a lack of specific evidence related to falls prevention interventions, there is evidence that older adults find mobile phones more usable than using a new device, for example a falls alarm.[14] It has also been suggested that barriers to smartphone use in this population can be overcome through adequate support and affordability.[15] There is evidence supporting the use of mobile phone-based healthy lifestyle programmes,[16 17] including to increase physical activity.[17–19] King et al[11] developed and tested smartphone applications (apps) based on behaviour change theory designed to motivate adults aged 45+years. One of these included personalised goal-setting and behavioural feedback, successful evidence-based behavioural change techniques.[20] The apps recieved positive feedback from participants and increased physical activity.

We know from previous studies that attitudes and beliefs are important to the uptake of and adherence to exercise by older adults.[21 22] The theory of planned behaviour (TPB)[23] is particularly useful for assessing older adults' attitudes in relation to exercise uptake and adherence.[21 22 24] The TPB is based on three core components:

1. Perceived behavioural control (PBC), the perceived ease or difficulty of performing the behaviour.
2. Social influences, including subjective norms (beliefs of important people, eg family), perceived social support (support from others for behaviour) and modelling (following observed behaviour of others).
3. Attitudes (outcome expectations).[23] Focused on the advantages and disadvantages of the behaviour (outcome expectations) and when related to adherence, whether these advantages have occurred.

Attitudes measured by using a TPB-based tool have been significantly associated with exercise behaviour in an earlier study.[21] This theory has informed the intervention overall and content of the motivational messages within the proposed intervention (focused on outcome expectations/PBC).

Smartphone technology-based motivational applications underpinned by behaviour change theory and developed with health professionals and older adults could be an effective way of encouraging maintenance of exercise and of successfully supporting adherence to evidence-based SBT. We have already carried out usability and acceptability testing of the technology and two motivational apps (one for health professionals and one for patients), before planning this trial. The smartphone apps have been developed through several cycles of user-led design. Initially we carried out engagement workshops with older adults (AgeUK) and health professionals from one falls service in Manchester, followed by usability/acceptability testing with another falls service in Manchester and their patients (Integrated Research Application System. IRAS:205980). The use of this approach has enabled us to develop the apps, establish whether the technology is acceptable to older adults and health professionals (qualitative methods) and to check its usability (technology testing). Overall, the apps were acceptable to both patients and health professionals with the majority of suggested changes made to the health professionals' app to ensure it fit more easily with their practice. Changes following this testing included improvements in the delivery of messages and a more streamline approach to scheduling activities for the health professional. Another suggested change was to make smartphone pens available to participants to aid in the use of the touchscreen.

This study now aims to explore whether it is feasible for smartphone technology to be used to support patients to sufficiently adhere to an evidence-based exercise rehabilitation programme. As a secondary aim it will assess whether technology-based outcome measures (smartphone-based falls alarm and timed up and go test (TUG))[25] are reliable when compared with standard methods (eg, falls calendars). Through a feasibility randomised controlled trial (RCT) we will explore the feasibility of using smartphone technology to support falls rehabilitation and test study procedures (eg, suitability of outcome measures, SD of the outcome measure, recruitment, randomisation, follow-up rates, retention, time required for analysis). Both arms of the trial will receive rehabilitation exercises and will report their exercises on a study provided smartphone but only the intervention arm will carry out goal setting and receive feedback through the phone.

The intervention has the potential to:

1. Increase the amount of support the patient receives to adhere to their exercise, leading to increased adherence.
2. Increase exercise progression/dose which could be cost neutral/saving.
3. Enable health professionals to monitor compliance to the prescribed programme.

This could assist maintenance of health, reducing long-term falls risk and re-access to services.

## METHODS

### Trial design

Core trial information is presented in table 1. This study is a two-arm pragmatic feasibility RCT including the

**Table 1** WHO trial registration data set

| Data category | Information |
| --- | --- |
| Primary registry and trial identifying number | ISRCTN:12 830 220 |
| Date of registration in primary registry | 21.08.2018 |
| Secondary identifying numbers | |
| Source of monetary or material support | National Institute for Health Research Postdoctoral Fellowship Award |
| Primary sponsor | University of Manchester |
| Secondary sponsor | N/A |
| Contact for public queries | Helen.hawley-hague@manchester.ac.uk |
| Contact for scientific queries | Helen.hawley-hague@manchester.ac.uk |
| Public title | The TOGETHER trial |
| Scientific title | Can smartphone technology be used to support an effective home exercise intervention to prevent falls among community dwelling older people? The TOGETHER feasibility RCT |
| Countries of recruitment | UK |
| Health condition of problem studied | Falls in older adults |
| Interventions | **Standard service:**<br>**Manchester City**: 12 weeks once a week contact (home or group exercise), check-ups until 6 months discharge.<br>**Trafford:** 8-week group exercise once a week or 6-week home exercise then discharged or referred to further 8-week group exercise.<br>**Control:** For all prescribed exercise plan and exercise booklet given, asked informally what they want to achieve (outcome goals).<br>Use of study provided smartphone for reporting exercises and falls detection as outcome measures only<br>**Intervention:** standard service (as above) plus the use of 'Motivate me' (health professional app) and 'My activity programme' (patient app) on study provided smartphones. |
| Key inclusion and exclusion criteria | **Age:** older adults aged 50+<br>**Sex:** male or female<br>**Inclusion:** at risk of falls, referred to falls rehabilitation services and assessed as suitable for an exercise programme, good 3G/4G reception in their home or wifi.<br>**Exclusion:** unable to follow instructions (unless they have support from a family member or carer), Severe visual impairment, long-term residential or nursing care, terminal illness or expected shortened lifespan, defined as less than 6 months, older adults unable to read written English unless they have support from a family member or carer). |
| Study type | Interventional<br>Allocation: randomised;<br>Primary purpose: prevention, feasibility |
| Date of first enrolment: | 20th September 2018 |
| Target sample size | 72 |
| Recruitment status | Pending |
| Primary outcome | Feasibility of the design and procedures |
| Key secondary outcomes | Balance (Berg), function (TUG/mTUG), falls (calendar/FallsMonitor@home), strength (30 s chair stand), fear of falling (Short FES-I), health-related quality of life (EQ-5D-5L/ ICE-CAP-O), resource use, adherence (my activity programme/EARS).<br>Baseline, 3, 6 months. |

EARS, Exercise Adherence Rating Scale; EQ-5D-5L, European Quality of Life 5 Dimensions; FES-I, Falls Efficacy Scale-International; ICE-CAP-O, ICEpop CAPability measure for Older people; ISRCTN, International Standard Randomised Controlled Trials Number; mTUG, Mobile based instrumented timed up and go test; TUG, timed up and go test.

collection of economic data. The trial design framework is exploratory. Alongside the trial, qualitative work is carried out to understand the feasibility of the intervention and the trial procedures.

## Sampling principles and procedures
### Eligibility
Older adults at risk of falls (aged 50+years) and assessed as requiring a falls rehabilitation exercise programme are identified through current community falls rehabilitation services delivered in Manchester City and Trafford. The two sites see patients from diverse socioeconomic populations. Exclusions include older adults who are unable to follow instructions (unless supported by a family member/carer), who are unable to understand written English (unless supported by a family member/carer), with severe visual impairment, those in long-term

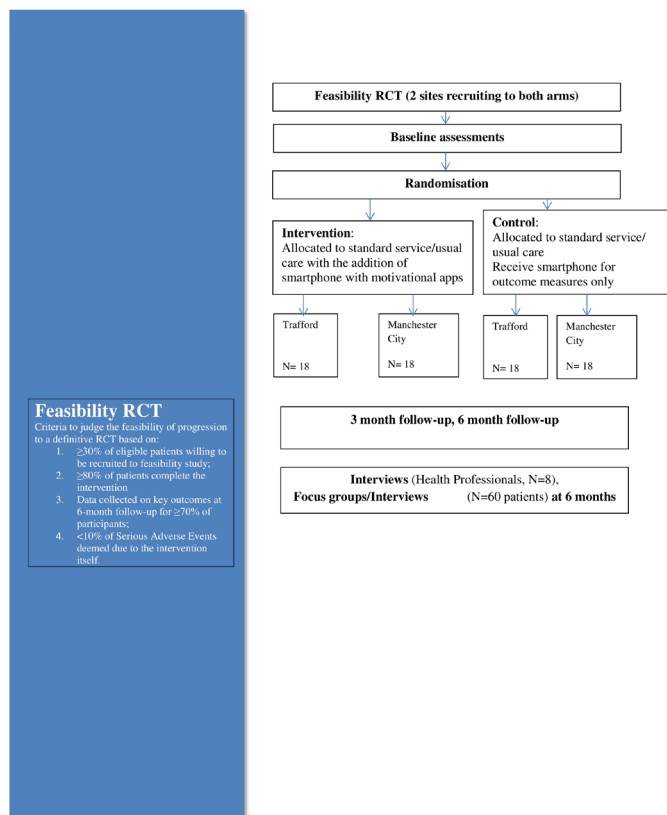

**Feasibility RCT (2 sites recruiting to both arms)**

↓

**Baseline assessments**

↓

**Randomisation**

↓

**Intervention:**
Allocated to standard service/usual care with the addition of smartphone with motivational apps

**Control:**
Allocated to standard service/usual care
Receive smartphone for outcome measures only

| Trafford | Manchester City | Trafford | Manchester City |
|---|---|---|---|
| N= 18 | N= 18 | N= 18 | N= 18 |

**3 month follow-up, 6 month follow-up**

**Interviews** (Health Professionals, N=8),
**Focus groups/Interviews** (N=60 patients) **at 6 months**

**Feasibility RCT**
Criteria to judge the feasibility of progression to a definitive RCT based on:
1. ≥30% of eligible patients willing to be recruited to feasibility study;
2. ≥80% of patients complete the intervention
3. Data collected on key outcomes at 6-month follow-up for ≥70% of participants;
4. <10% of Serious Adverse Events deemed due to the intervention itself.

**Figure 1** Consort diagram.

residential or nursing care and those with terminal illness or expected shortened lifespan, defined as less than 6 months, as determined by the NHS teams. Patients need to have good 3G/4G mobile phone reception (able to access webpages) or wifi in their home and this is assessed by the health professional before they handout participant information or by the researcher when taking consent.

### Recruitment, consent, sample size

Health professionals give patients the study information sheet and inform them about the intervention. The health professionals then ask the patient if they are happy to be contacted by the researcher who demonstrates the technology either in the patient's home or within groups at each NHS site. The technology is demonstrated to the participant before they are asked to give informed consent. Where possible a former patient who has used the smartphone applications accompanies the researcher to demonstrate the technology. We think involvement of a peer has the potential to assist in promoting patient confidence in the use of the technology.

The first 36 eligible patients identified through each service (n=72 in total) who are willing to participate are being recruited and randomised (figure 1). Thirty patients per arm after attrition (approx. 10%) are normally used for feasibility RCTs.[26] Study participants are randomised using a computer-generated randomisation algorithm at sealedenvelope.com, stratified by gender and site, using block randomisation (2, 4, 6 blocks) into either

intervention or control group. Stratification is by gender and site to ensure equal distribution across sites as we are testing all trial procedures.

### Blinding

Baseline and follow-up (3 and 6 months) assessments are carried out by experienced clinicians within each NHS Trust (not a member of the clinical teams participating), who are blinded to which intervention the participants are receiving; at baseline the individual is also blinded to the intervention they will receive as randomisation occurs after baseline assessment.

As this is an 'active' intervention, it is not possible to blind the health professionals delivering the service or the participants during the intervention. The lead researcher provides technical support to both arms to use the smartphone so is not blinded.

The statistical analysis will be carried out by the lead researcher with the support of a statistician. Patient ID codes will be removed from the data to allow for blinded analysis.

### Patient withdrawal

In consenting to the trial, patients are consenting to the trial treatment, follow-up and data collection. If withdrawal of the randomly allocated treatment occurs, patients should still be followed up where they agree. Patients are allowed to withdraw without giving reason at any time and a withdrawal case report form (CRF) will be completed to document the date and reason (where given) for withdrawal. Data collected up to the time of withdrawal will be included in analyses. Health professionals will assess patients' capacity to take part in the rehabilitation programme and the study; if they have been deemed to have lost capacity to consent they will be withdrawn from the study but the data already collected will be retained.

### Interviews with patients

All participants are offered an interview (even those who withdraw from the trial) in their own home after the final follow-up to assess their experiences of the intervention and trial processes. Family members/carers may also attend the interview at the participants' request.

### Focus groups with health professionals

Health professionals from Trafford and Manchester City who are involved in the study are recruited to participate in a focus group at the end of the study (after 24-week follow-up). All members of staff (n=8) will be given study information by their team leader and asked if they are available for a focus group; the focus groups will take part at their place of work at a time convenient to each team. Participating staff can choose to be part of a one-to-one interview if they prefer not to be interviewed with colleagues or if for staffing reasons it is not feasible for them to attend the focus group.

## The intervention

Full details of the intervention components are shown in online supplementary material: table 1 (Template for Intervention Description and Replication (TIDieR) Guidelines).

### The technology

The Samsung Galaxy J5 as a means of communication[27] will be provided to all participants and health professionals. Samsung phones have been used previously in our research, with good usability and have the correct specification for the falls detector to work.[10] The research team will provide technical support for participants and health professionals (HH-H) and any required application updates (SM, CTa).

### 'Motivate me' app

The 'Motivate me' app is the health professional application. This app is used by the health professional with the patient to set behavioural/outcome-based goals, for the health professional to see what exercises the patient has reported and to give feedback and to check they have received messages (online supplementary figure 1).

### 'My activity programme'

'My activity programme' is the patients application. This app will be used by the patient to report the exercises they have done, receive messages and prompts and to confirm whether they like the messages received (online supplementary figure 2).

There are 12-behaviour change techniques adopted[28] through the intervention include goal setting (behaviour/outcome), action planning (recording plan to exercise in diary on smartphone/reminder text messages when it is time to start the programme) and feedback on behaviour (providing feedback on what they have done/benefits). The key behavioural change techniques delivered as part of the control and intervention arms are outlined in table 2.

### The control

Standard service is variable across different sites, but all sites deliver a mix of the evidence-based Falls Management Programme and Otago[29] exercises as standard care.

They include face-to-face delivery once-a-week and a prescribed home-exercise programme (with booklet) with informal outcome-based goal setting.

Manchester City: Once a week visits (either home-based or group exercise) for around 12 weeks (dependent on need) and then check-ups until 6 months discharge.

Trafford: 8-week group exercise once a week then discharged, or 6-week home-based exercise then discharged or referred to further 8-week group exercise.

Both sites leave participants with a home exercise plan on discharge and where appropriate refer onto community-based strength and balance programmes.

| Table 2 Behaviour change techniques adopted* | | | | |
|---|---|---|---|---|
| | **1.Intervention arm** | **1a How** | **2.Control arm (standard service)** | **2a How** |
| 1.1 Goal setting (behaviour) | x | What, when, where—smartphone and paper | x | What, Where—paper |
| 1.3 Goal setting (outcome) | x | Smartphone verbally | x | Verbally |
| 1.4 Action planning | x | Smartphone | | |
| 1.5 Review behavioural goals | x | Smartphone verbally | x | Paper verbally |
| 1.7 Review outcome goals | x | Smartphone verbally | x | Verbally |
| 2.2 Feedback on behaviour | x | Smartphone verbally | x | Verbally |
| 4.1 Instructions on how to perform the behaviour | x | Physically smartphone paper | x | Physically Paper |
| 5.1 Information about health consequences | x | Smartphone verbally (ad hoc) | x | Verbally (ad hoc) |
| 5.6 Emotional consequences | x | Smartphones verbally (ad hoc) | x | Verbally (ad hoc) |
| 6.1 Demonstration of behaviour | x | Physically | x | Physically |
| 7.1 Prompts | x | Smartphone | | |
| 8.7 Graded tasks | x | Smartphone paper | x | Paper |

*Based on Michie et al's[28] behaviour change taxonomy.

## Control application for self-reporting exercise

The control arm receives a study phone with a basic app where they report their exercises, but they are only able to report their exercises (outcome measure), they are not able to view their programme, receive messages or receive feedback on the phone. The health professional is not able to view what they have reported (outcome measure for the research team only), thereby minimising risk of contamination.

## Co-Treatments

Trial participants are free to seek management of falls and other related or unrelated medical conditions during the course of the trial. We record all health service resource use and these will be reported as a trial outcome. At trial closure, participants will continue with usual healthcare; no further ancillary care is provided beyond that immediately required for the proper and safe conduct of the trial.

**Table 3** Schedule of enrolment interventions and assessments

| | Study period | | | | | Post-intervention |
|---|---|---|---|---|---|---|
| | **Enrolment** | **Allocation** | **Post-allocation** | | | |
| Timepoint | $-t_1$ | **0** | $T_1$ | $T_2$ | $T_3$ | |
| **Enrolment** | | | | | | |
| Eligibility screen | X | | | | | |
| Consent to further information | X | | | | | |
| Tech demo and Informed consent | X | | | | | |
| Allocation | | X | | | | |
| **Interventions:** | | | | | | |
| *Control:* *Manchester City* **Trafford** | | | ●—————————————● ●————————● | | | |
| *Intervention* | | | ●—————————————● | | | |
| **Assessments:** | | | | | | |
| Gender Age Ethnicity Education Housing Falls history Medical history Previous mobile/smartphone use Allocated to home or group exercise | X | | | | | |
| *Falls (calendar)* *Falls (alarm)* *My activity self-report* *Prescribed exercise plan* **Face-to-face delivery** | | | ●—————————————● | | | |
| *Berg* *TUG mTUG* *30 s Chair stand* *FES-1* *EQ5D* *Resource use* **Health professional time resource** *ICE-CAP-O* **EARS** | X | | | X | X | |
| *Interviews* **Focus groups** | | | | | | X |

EARS, Exercise Adherence Rating Scale; EQ5D, EuroQol five dimension scale; FES, Falls Efficacy Scale; ICE-CAP-O, ICEpop CAPability measure for Older people; mTUG, Mobile based instrumented Timed up and go test; TUG, Timed up and go test.

## Outcome measures

Data such as demographics (age, gender, socio-economics, health conditions, falls history, previous smartphone/mobile phone use and wifi) and physical tests are recorded on the CRF (table 3).

### Primary outcome measures

Assess feasibility and acceptability of the design and procedures including:

1. Willingness of participants to be randomised (collected on the screening section of the CRF/interviews).
2. Willingness of clinicians to recruit participants (through focus group/completion of screening on the CRF).
3. Number of eligible patients (through the CRF and documentation of number of monthly referrals to falls services).
4. Whether demonstration by peer of the technology aids recruitment.
5. Characteristics of the proposed outcome measures, for example reliability of falls detector when compared with falls calendars, whether a self-report app is a reliable outcome measure.
6. Follow-up rates, adherence/compliance rates.
7. Time needed to collect and analyse data.
8. Determine effect sizes for use in sample-size calculations, enabling power calculations for the reduction in falls for a definitive large-scale RCT.

Start/stop criteria for going to full trial are included in Consolidated Standards of Reporting Trials (CONSORT) diagram (diagram 1) based on these outcome measures.

We will also report intervention fidelity, process and compliance using observation during quality assurance visits. Health professionals and the assessors will follow a standard operating procedure for assessment and intervention. The trial treatment record (CRF) includes details of the grade and type of staff involved with delivery. The health professionals and researcher will keep an issue log of technical issues with the phones or apps during the trial either experienced directly or reported by participants. We will also explore the potential impact of differing length of exercise delivery across sites.

### Outcome measures
#### Falls

The primary outcome for any future definitive trial would be falls, expressed as fall rate per person per months of follow-up observation after randomisation. This study collects falls data for the purposes of testing feasibility of data collection, and to inform us of falls rates and intervention effect size for a future sample size calculation.

All participants will wear the smartphone in their pocket or on a waistband and this will act as a falls detector, running the FallsMonitor@home app developed by the University of Bologna.[10] If the patient chooses then they can also use the app on the phone as a falls alarm. The fall detection system application allows the user to identify a list of formal/informal caregivers who will receive an SMS if a fall is detected. Patients are given an opportunity to de-active the falls alarm through an application on the smartphone if there is a false alarm, enabling the user to maintain control and prevent unwanted intrusion. Participants are asked if we can use their anonymised falls data for further development of the app and in the FARSEEING real-world falls database.[30]

To validate this as an outcome measure we use the internationally agreed Prevention of Falls Network Europe (ProFaNE) falls definition[31] and follow the agreed ProFaNE falls data collection and analysis protocols based on self-report calendars.[32]

#### Fear of falling

Short Falls Efficacy Scale-International (Short FES-I) is used to measure fear of falling.[33] This is often a measure used by UK falls services as part of standard outcome measures.

#### Function

The TUG will be used to assess improvements in mobility and function. The TUG will be applied as described by Podsiadlo and Richardson.[25] Participants will be asked to perform the TUG at their self-selected habitual walking speed. A medical device implementing an instrumented version of the TUG will be used (mTUG, mHealth Technologies). The device is able to automatically provide guidance to the user for administering the test, capture and process the data, and generate summary reports of function for the health professional. The blinded assessor will complete the normal TUG and the mTUG as outcome measures (the standard TUG as a validation measure) to explore whether the mTUG is usable as an outcome measurement for the definitive RCT. The health professional will carry out the mTUG with a sub-sample of 10 patients at each site to assess their experiences of its use.

#### Balance

The Berg Balance Scale will be used to assess balance. This has good validity and sensitivity in this population[34] and is one of the best outcome measures for assessing standing balance.[35] It has also been used for the prediction of falls.[36] The effect sizes from this outcome measure scale will be used as part of the power calculation for the full trial.

#### Strength

30 s Chair stand test,[37] which has good validity and is used throughout health services, will be used to assess physical ability, in particular strength.

### Adherence

Adherence will be measured in a number of ways (outlined in detail, table 4):

1. Self-report app will be used for both control and intervention groups. Adherence will be classed as the participant carrying out 80% of their prescribed programme (based on the evidence base for effective strength and balance).[6 38]

**Table 4** Adherence measures

| | What | How/additional validation |
|---|---|---|
| Self-report through my activity programme and control arm smartphone app | Exercises reported on app to their prescribed programme the day they are carried out.<br>► Exercise type<br>► Intensity<br>► Dose<br>Adherence defined as participant carrying out 80% of their prescribed programme. | The health professional will be asked to provide a copy of the participants prescribed exercise plan, any changes to it and the dates any changes were made (both sites record this as part of standard intervention).<br>For face to face home delivery, the health professional will be asked to report exactly what the patient has done when with them (this will be used to validate the self-report from participants).<br>After discharge from rehabilitation if participants move onto other strength and balance provision. Those services will give us copies of the exercise programme delivered for any days the participants attend, attendance records and any prescribed home exercise programme. |
| EARS | Validated 16-question tool with a 6-question subscale specifically measuring adherence. | Paper questionnaire at baseline, 3 months and 6 months. |

EARS, Exercise Adherence Rating Scale.

2. Exercise Adherence Rating Scale (EARS).[39] This is a validated 16-question tool with a 6-question subscale specifically measuring adherence (remaining questions measure reasons for adherence/non-adherence).

### Health economics

The health-related quality of life measures will include the European Quality of Life 5 Dimensions (EQ-5D-5L)[40] and an additional measure used in previous trials related to falls prevention (the ICEpop CAPability measure for Older people (ICE-CAP-O).[41 42] Costs of delivering the intervention will be observed based on staff training, delivery costs and equipment costs. Additional resource use measures will be captured via a Resource Use Questionnaire which will seek to measure costs related to an NHS and social care perspective (secondary, primary, community care service use), and a patient perspective (costs related to informal care). The findings from these will inform the feasibility of collection of the data, and priorities for cost collection at full trial.

### Interviews/focus groups

The interview and focus group schedules are based on FARSEEING guidelines.[43] The following key areas will be explored in relation to the smartphone, the 'Motivate me' app, 'My activity programme' app, FallsMonitor@home and the mTUG. Ease of use, clarity of screen, demonstration of use, wearing comfort, adaption of use, reliability, choice and control and home and lifestyle. This feedback will be considered and any required changes to the technology set-up, applications and intervention will be made prior to the definitive RCT. We also ask additional questions about the research process including general expectations and views; experiences of recruiting patients (health professional), and of being recruited and randomised (patients), suggestions of methods for recruiting participants; likely uptake and retention of participants.

### Analysis

Quantitative data are analysed using SPSS Release V.22.0. The main analyses is descriptive, involving the estimation of recruitment rates, attrition rates, non-compliance rates, means and SD of outcomes by group at baseline and end trial, and 95% CI for differences of means of outcomes between groups and assessment of change following the intervention at end trial. The health economics analysis is focused on informing relevant measures and means of collection of health-related quality of life and resource use for the future definitive study. Only an exploratory cost-effectiveness analysis will be conducted; for all measures we will report mean values and sample variability alongside information on missing values.

Data from the smartphone-based outcome measures (FallsMonitor@home, mTUG, My activity programme/control self-report app) will be compared with the traditional measures (falls calendar,[32] TUG,[25] EARS[39]) alongside qualitative feedback as part of their validation. A statistical analysis plan will be created before data analysis.

Qualitative interviews/focus groups will be analysed using thematic analysis.[44] The research will be inductive and although will seek to further understand the quantitative findings, this approach will also generate categories and explanations directly from the data rather than based on previously set aims and objectives, reducing

risk of bias.[44] QSR International's NVivo V.10 qualitative data analysis software will be used to manage the data. The validity of the analysis will be checked by returning to the data once themes have been identified and also through the use of a second researcher who will check samples of analysis. The accuracy of the transcripts will be checked through discussion with participants to establish if anything is not clear from the interviews/focus groups.

### Ethical issues

Regional and site-specific approvals have been obtained. We are collecting and storing personal information in accordance with the General Data Protection Regulation and Data Protection Act 2018. As this is a study with older patients a number of ethical issues could arise. To address these, community services will act as gatekeepers to access patients and assess patients' eligibility for the study. The intervention is delivered by health service staff and provided in addition to standard service; therefore patients are unlikely to be disadvantaged.

If falls are detected by the smartphone, it is important that someone is informed in real time. The smartphone application allows the user to select a list of formal/informal caregivers who will receive an SMS if a fall is detected. It will be made clear that the falls service is not an emergency service so in the event of a fall the person receiving the text message would call an ambulance as they would in normal circumstances. If patients already wear a call alarm then they will be encouraged to continue to use this as well or to adopt their usual method of alerting help.

The study requires monitoring of subjects and it is important that patients do not find this obtrusive (privacy issues have been identified as major barriers to the use of technology). Patients are given an opportunity to de-active the falls alarm through an application on the smartphone if there is a false alarm. However, previous consultation/usability testing with older adults raised no major privacy issues.

There are ethical issues in the removal of technology at the end of studies.[45] We will not be able to offer older adults the technology at the end of the 6-month study period, but they will be offered the opportunity to download the apps onto their own phones if they wish.

The risk of interviews and focus groups are minimal. The patient or health professional can ask the researcher to move onto another question if they are uncomfortable at any point. Health professionals will be given the chance to discuss the trial, technology and intervention in a one-to-one interview if they do not feel comfortable giving feedback in front of colleagues.

### Patient and public involvement

Patient and public representatives have been involved in designing the trial including outcome measures. Feedback from previous usability testing with patients and from patients who sit on our Advisory Group (AG) provided direct information on the design of the trial for example, use of self-report app for control arm. Patients on our AG (who were formerly patients of one of the services) helped to design study material such as the patient information sheet. They assisted in training health professionals in approaching patients for recruitment and goal setting as part of the intervention. Three participants, who took part in our usability testing, became peer mentor volunteers for the trial. They will attend the first visit (if the patient gives permission) to demonstrate the technology to patients before consent is given. We will explore whether peer involvement aids recruitment. Finally, the volunteers and the patients who sit on our AG will aid with dissemination of study findings, for example helping to arrange dissemination events and providing feedback on newsletters for participants.

### Trial monitoring

The lead researcher (HH-H) will monitor the delivery of the intervention and recruitment of patients; there will also be a clinical lead (AE, EM) at each site taking overall responsibility for identification of patients and delivery of the intervention. This team, alongside academic experts (JLH, LC, SM, ASM, CTo) from the Trial Co-ordination Group will ensure overall quality of trial data. The AG meets bi-annually, giving feedback on the project, providing expert guidance and assisting in dissemination; this includes two previous patients. A risk register is reviewed by the AG. The study is subject to the audit and monitoring regime of the University of Manchester and the monitoring plan followed.

A detailed risk assessment has been carried out and potential patient, organisational and study hazards considered, the likelihood of their occurrence and the resulting impact should they occur.

### ADVERSE EVENTS

A safety reporting protocol has been developed for related and unexpected serious adverse events (AEs) and directly attributable AEs. An AE is defined as any untoward medical occurrence in a subject which does not necessarily have a causal relationship with treatment. These AEs are recorded in the CRF and if a serious AE occurs then it is reported to the chief investigator (CI). The CI will determine whether AEs require reporting to the trial sponsor and ethics committee, in accordance with the safety reporting protocol.

### DISCUSSION

This is the first trial that we are aware of that explores the potential use of motivational smartphone apps for the support of an evidence-based falls exercise programme.

As this is an active intervention and control we are unable to blind participants or those delivering the intervention. However, the design does enable us to blind those carrying out both the assessments and analysis. The fact that both arms have a smartphone minimises the risk of unblinding with the independent assessors and, we

would argue, also reduces risk of drop-out. There is the potential for the control group to become motivated by reporting their activities. However, if we did not ask them to report, there is also the risk of any difference between groups being a function of differential reporting schedules rather than a function of the intervention per se.

We provide participants with study phones, which may be different to using the app on their own phones. However, we need to ensure the smartphone meets the technical specification required for FallsMonitor@home to work correctly. Furthermore use of study phones enables us to maintain confidentiality of participants (if phones are lost we can wipe them remotely).

This trial assesses several novel outcome measures against the gold standard, the mTUG against standard TUG, the FallsMonitor@home against standard calendar method and a self-report app against the EARS tool.[39] This enables us to further our understanding of whether technology has the potential to provide more objective and reliable outcome measures than current methods.

We use two very different NHS sites, reflecting the reality of day-to-day practice (one specialist falls service, one general rehabilitation services) to explore the delivery of the intervention. This means that the standard service is different across the two sites adding complexity to how the control and intervention arm are delivered. However, these differences enable us to assess its scalability to full trial where different types of falls services would need to be included as sites. It also enables us to be more representative of current services and assess its potential for delivery in practice.

**Author affiliations**
[1]Division of Nursing, Midwifery and Social Work, School of Health Sciences, Faculty of Biology, Medicine and Health and Manchester Academic Health Sciences Centre, University of Manchester, Manchester, UK
[2]Health Sciences and Technologies-Interdepartmental Center for Industrial Research, University of Bologna, Bologna, Italy
[3]mHealth Technologies srl, Bologna, Italy
[4]Department of Electrical, Electronic and Information Engineering 'Guglielmo Marconi', University of Bologna, Bologna, Italy
[5]Central Manchester University Hospitals NHS Foundation Trust, Manchester, UK
[6]Pennine Care NHS Foundation Trust, Ashton-under-Lyne, UK
[7]Centre for Health Economics, University of York, York, UK
[8]Division of Dentistry, School of Medical Sciences, Faculty of Biology, Medicine and Health, University of Manchester, Manchester, UK
[9]Department of Clinical Gerontology, Robert-Bosch-Krankenhaus, Stuttgart, Germany
[10]Department of Neuromedicine and Movement Science, Faculty of Medicine and Health Sciences, Norwegian University of Science and Technology, Trondheim, Norway

**Acknowledgements** We thank Professor Dawn Skelton and Later Life Training for allowing us to use their images and the name 'Motivate Me' for the health professional app. We thank two Public and Patient Involvement (PPI) representatives who sit on our Advisory Group and three peer volunteers who are supporting recruitment.

**Contributors** HH-H leads the research project and its design, managing the trial overall and has led the writing of the protocol. CTa and SM give technical support for the study and have advised on outcomes and the manuscript. JLH, LC, ASM, CTo and SM have provided scientific advice around the design of the study and commented on the manuscript. T-LS and FBY have given advice on statistics and health economic part of design and manuscript. AE and EM have given advice on the operationalisation of the study and commented on the manuscript.

**Funding** Dr Helen Hawley-Hague is funded by a National Institute for Health Research (NIHR) (Postdoctoral Research Fellow) for this research project. This publication presents independent research funded by the NIHR. The views expressed are those of the author(s) and not necessarily those of the National Health Service, the NIHR or the Department of Health and Social Care.

**Competing interests** None declared.

**Patient consent for publication** Not required.

**Provenance and peer review** Not commissioned; externally peer reviewed.

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
