## [Reviewer comments · BMJ Open]

ARTICLE DETAILS

TITLE (PROVISIONAL)	Can smartphone TechnolOGy be used to support an EffecTive Home ExeRcise intervention to prevent falls amongst community dwelling older adults? The TOGETHER feasibility RCT study protocol.
AUTHORS	Hawley-Hague, Helen; Tacconi, Carlo; Mellone, Sabato; Martinez, Ellen; Easdon, Angela; Yang, Fan; Su, Ting-Li; Mikolaizak, A. Stefanie; Chiari, Lorenzo; Helbostad, Jorunn; Todd, Chris

VERSION 1 – REVIEW

REVIEWER	Jacob Sosnoff Department of Kinesiology and Community Health College of Applied Health Sciences University of Illinois at Urbana-Champaign Urbana, Illinois, United States of America Ownership in Sosnoff Technologies, LLC
REVIEW RETURNED	04-Dec-2018

GENERAL COMMENTS	The protocol outlines an interesting use of smartphone technology to maximize adherence to exercise program designed to prevent falls. The researchers may want to quantify participants familiarity with smartphone as this may be a confound.
---

REVIEWER	Julie Bruce Warwick Clinical Trials Unit, University of Warwick, UK I know some of the research team, lead author and Prof Todd from the falls community and conferences although have not worked together.
REVIEW RETURNED	07-Dec-2018

GENERAL COMMENTS	This is an interesting protocol from a group of experts who are well-known and respected in the field of falls prevention. This is an important topic and worthy research question, thus how best to encourage and promote adherence to exercise interventions in those older adults who have already fallen or who are at high risk of falls. I have a number of comments and these are made in the spirit of improving the protocol rather than criticisms. It is unclear to what extent these can be addressed given that the first participant was recruited in September and the study has therefore, already started.
--

Firstly, the design. This is a feasibility study and yet there seems to be confusion in the description throughout the manuscript - it switches between a full, definitive RCT design and includes elements and planned analysis for a definitive RCT rather than sticking to the aims and objectives of an exploratory, feasibility study. It needs to be much clearer whether this is feasibility work or a pilot RCT (see NIHR guidance). Currently it is a mix of both 'can this be done' (feasibility) and a pilot (e.g. main RCT run in miniature/internal pilot).

Abstract

Under methods, states this is a 2-arm 'simple' RCT rather than feasibility. Outcomes are feasibility and acceptability but the secondary outcomes are all trial-related outcomes including resource use. No mention of providing participants with a smartphone, only an motivational 'app'. I assumed this was available to download for those who had a smartphone already & that having a smartphone would be an eligibility requirement. Under strengths & limitations, it states that results will be directly applicable to practice rather than feeding into a larger definitive RCT (doesn't match what it states later). You would not want the findings from this study to filter into practice without further testing within a larger RCT. Also it states here that the study is complex, despite being described several lines earlier as a 'simple' RCT.

The background section is very clear and justifies the need for finding solutions to encourage adherence to exercise interventions. Page 4, line 36/7 introduces the term 'apps' but then elaborates in full, referring to smartphone motivational applications on page 5.

On page 5, lines 11-18 explain that usability and acceptability testing have already been completed with older adults and health professionals, yet this is the purpose of this study (based on the abstract?). So this was a bit confusing as there is an ISRCTN number added, but this relates to the current protocol rather than a previous study? Is that correct?

Line 24 states that a secondary aim is to assess whether technology based outcome measures are reliable (compared to what?). How will the reliability assessment be undertaken?

Methods

Trial design - now incorporates economic analysis which is not a feature of a feasibility studies? you are not trying to determine cost-effectiveness at this stage and it does state that yet you are collecting HRQoL, healthcare resource use etc. and mentioning economic analyses.

Table 1 does not have the trial registration number.

Table 1 does not clearly specify that the study intervention is a smartphone with an application. Surely the trial intervention(s) is provision of a smartphone with goal-setting versus provision of a smartphone without goal-setting? The description of the falls prevention service is irrelevant as it is identical in both arms. Are people actually provided with a smartphone - that wasn't clear to the reader until page 9 where it describes the control arm receive will also receive a Samsung Galaxy with an app? Also healthcare professionals receive a smartphone? What about contamination here - has that been considered?

	It is not mentioned under the intervention in the flowchart diagram hence difficult for the reader to follow what is being given as the intervention/control. The control states allocated to usual care - but they are given a smartphone with an app thus it is comparing 2 active interventions and there is no "usual care" comparison group? Under study type it is described as a phase II clinical trial. Under design, unclear why block randomisation is being used for 30 patients and stratification by age, site when it is feasibility unless the data from this study are feeding into a definitive RCT as internal pilot data. Under page 11, intervention fidelity checks and process evaluation - this is actually an outcome of your feasibility study. Again, written as a definitive RCT in sections which is confusing. Primary outcomes "characteristics of proposed outcome measures e.g. reliability of falls detector" - how is this being determined? Under secondary outcomes, it describes that the smartphone will be on a waistband and will act as a falls detector. This is not described under the intervention section - thus it is a specialised device rather than a phone. Unclear how the falls app will be compared to self-report falls data. Will both arms have this facility? Seems to be another sub-study as well with another medical device for the TUG - that's fine but these should be clearly stated in the aims and objectives section rather than introduced within the outcomes measurement. Health economics Why record participant healthcare resource use etc. when the purpose is to work out how much additional costs are required for training in the use of this new technology. Similarly under the analysis section, why are you reporting an intention to treat approach when this is feasibility work, also adjusting for baseline characteristics - highly unusual for feasibility as you don't have a large enough sample for regression analyses? I have been fairly critical although am very supportive of the study and the concept, the description of the design and methods could be improved. The team are experts in falls prevention but further clarity would improve the reporting of this study protocol. Good luck with this important study.
--	--

REVIEWER	Dr Alice Richardson Australian National University Australia
REVIEW RETURNED	11-Feb-2019

GENERAL COMMENTS	In this paper, the authors have presented the protocol for a research project to study the feasibility of smartphone technology to prevent falls in older people. Three items from the Review Checklist (Q9 – Q11) are not applicable in my view because the paper is a protocol not a full study.
--

	Below I will elaborate on the one item in the Review Checklist where my response was “No”. My concerns are of a relatively minor nature and so I recommend that the paper undergo minor revision before resubmission. Q7: the statistics are dealt with in a somewhat conflicting manner. Firstly, the abstract states that the analyses will be “primarily descriptive” and yet the analysis section (page 15 – 16) mentions a number of inferential methods. The authors do note that the inference will be treated with “great caution” but they could state this before the analysis methods are listed, not after. Secondly, the section on the economic analysis on page 14 seems underdeveloped in comparison to the naming of several instruments to measure quality of life and so on. The authors could be more specific about the tools planned for the economic analysis. Thirdly, I am concerned that the primary outcome will be too rare in the sample size for any meaningful descriptive statistics to be calculated. The authors could give an indication from previous research what the typical fall rate in community-dwelling older persons is to satisfy the reader that a sample size of about 70 will be sufficient for the outcome measures planned. Thankyou for the opportunity to be part of the academic referring system in this way.
--	---

REVIEWER	Edward Meinert University of Oxford United Kingdom
REVIEW RETURNED	11-May-2019

GENERAL COMMENTS	The authors present a novel study to investigate the feasibility of using a digital health app to support adherence to fall rehabilitation/strengthening programmes. As the authors note, the use of these technologies can serve as a mechanism to provide scalable delivery of monitoring and adherence, leading to better outcomes. The challenge in this type of research is that the evidence on how to best implement these technologies is limited. Additionally, there is a tendency to focus first on the utility of the technology and not enough emphasis on first considering user needs. The authors have taken care to test and iterate in the construction of their intervention and now seek to do more rigorous testing on feasibility. Critical review comments herein are centred on providing detail as to the intervention design and subsequent details on the study approach; I am confident the authors have such detail to hand and will be able to enhance the manuscript to reflect their efforts to date and sets a strong foundation for the study. 1. Page 5, rows 25 through 28 - what is delivered by smartphones to justify the use of this technology in this population? What barriers from the literature have the authors identified and what differences could the target population have in their use of this technology which could make adoption challenging? While smartphones could be the solution, is there sufficient evidence to suggest that there is enough saturation of smartphone use in this demographic to justify the cost/development of such technologies for this purpose? What alternates have been considered? 2. Page 6, rows 8 through 32 - what was the result of your previous usability and acceptability resting? What did you learn
--

	from this work and how did this impact subsequent application development and system implementation? 3. Overall comment (introduction/background) - it would be useful to have figures illustrating the overall system enterprise architecture and an overview of the system functionality/user interface. Though I do note you speak about the intervention later in the manuscript. It is ok to do this there, but the same comment applies as the details provided in the Intervention section do not provide detail to what has been developed, nor the way the system has improved over time via feedback to inform the current state of the study. 4. Page 7, row 42 (interventions) - what impact does the variant durations have on results? What is the length of the intervention use including the digital health app intervention? 5. Page 8, row 9 (Inclusion and exclusion criteria) - how has sampling been completed to ensure demographic saturation (e.g. education, socio-economic, ethnicity, etc.) 6. Page 8, rows 48 through 59 (eligibility) - how have the authors taken into account the difference in data performance from 3G to 4G to Wifi. How will this be assessed and what criteria used to define "good"? 7. Page 9, row 13 and 14 - what is the reference for the point regarding peer work and it is unclear how peer work will be used in the study or is a variable which is monitored. 8. Page 10, row 22 - why was this particular smartphone selected for the study? 9. Page 10, row 28 - who developed the motivate me app and how is continuous application development and support for the technology administrated and planned throughout the study? 10. Page 12 (outcome measures) - what will happen if you do not receive demographic saturation of participant types? 11. Page 14, row 33 - how will you ensure the participants use the smartphones as you envisaged? Is it realistic to think that participants will always use a smartphone in this manner; do you have previous evidence from your previous work that indicates the likelihood of the success of this? 12. Page 17, row 57 - you are offering participants to use a new mobile phone that is not their pre-existing mobile phone; how have you accounted for the difference in the functionality of potentially more sophisticated devices than their current devices? 13. It would be useful to see additional references to recent smartphone research which will have extensibility to this study both in study design and intervention design.
--	--

VERSION 1 – AUTHOR RESPONSE

Reviewer: 1

Please leave your comments for the authors below The protocol outlines an interesting use of smartphone technology to maximize adherence to exercise program designed to prevent falls. The researchers may want to quantify participants familiarity with smartphone as this may be a confound.

Thank you- we have made this clearer in the text (see bottom of page 11). We designed the app with older adults who had never used smartphones before and provided them with phones so therefore it was not an exclusion criteria. This is a pragmatic trial and we wanted to test it with as many of the patients who undergo rehabilitation as possible so as to make it transferable to practice. However, we

have collected data on whether they own a smartphone, a mobile phone or have wifi so this can be explored. We have made this clearer on (page 12 and Table 3).

Reviewer: 2

Reviewer Name: Julie Bruce

Institution and Country: Warwick Clinical Trials Unit, University of Warwick, UK Please state any competing interests or state 'None declared': I know some of the research team, lead author and Prof Todd from the falls community and conferences although have not worked together.

Please leave your comments for the authors below This is an interesting protocol from a group of experts who are well-known and respected in the field of falls prevention. This is an important topic and worthy research question, thus how best to encourage and promote adherence to exercise interventions in those older adults who have already fallen or who are at high risk of falls.

I have a number of comments and these are made in the spirit of improving the protocol rather than criticisms. It is unclear to what extent these can be addressed given that the first participant was recruited in September and the study has therefore, already started.

Firstly, the design. This is a feasibility study and yet there seems to be confusion in the description throughout the manuscript - it switches between a full, definitive RCT design and includes elements and planned analysis for a definitive RCT rather than sticking to the aims and objectives of an exploratory, feasibility study. It needs to be much clearer whether this is feasibility work or a pilot RCT (see NIHR guidance). Currently it is a mix of both 'can this be done' (feasibility) and a pilot (e.g. main RCT run in miniature/internal pilot).

Thank you for your feedback. This is a feasibility study and not a pilot and we have gone through the manuscript carefully and made some amendments to ensure this is reflected.

Abstract

Under methods, states this is a 2-arm 'simple' RCT rather than feasibility. Outcomes are feasibility and acceptability but the secondary outcomes are all trial-related outcomes including resource use.

This has been amended to state it is a feasibility RCT. The secondary outcomes have been amended to state that we are exploring standard measures against instrumented versions.

No mention of providing participants with a smartphone, only a motivational 'app'. I assumed this was available to download for those who had a smartphone already & that having a smartphone would be an eligibility requirement.

Apologies, in fact both arms are given a phone, the control arm for outcome measures only and the intervention arm to receive the intervention. This was for data security, standardisation and to enable the falls detector/alarm to work (requires a phone of a certain specification). This has been made clearer in the abstract.

Under strengths & limitations, it states that results will be directly applicable to practice rather than feeding into a larger definitive RCT (doesn't match what it states later). You would not want the findings from this study to filter into practice without further testing within a larger RCT. Also the states here that the study is complex, despite being described several lines earlier as a 'simple' RCT.

Thank you for this comment. This has been made clearer, as the app is being tested directly with the clinical teams then it means whether the clinical teams and patients can use it is directly applicable to

practice in terms of acceptability and feasibility of the intervention. It of course does not tell us whether it is effective. This has been clarified. The line stating the design is complex has been removed as it is covered by the line stating it is a pragmatic feasibility trial.

The background section is very clear and justifies the need for finding solutions to encourage adherence to exercise interventions. Page 4, line 36/7 introduces the term 'apps' but then elaborates in full, referring to smartphone motivational applications on page 5.

This has been amended using smartphone applications the first time with apps in brackets and then using apps following this.

On page 5, lines 11-18 explain that usability and acceptability testing have already been completed with older adults and health professionals, yet this is the purpose of this study (based on the abstract?). So this was a bit confusing as there is an ISRCTN number added, but this relates to the current protocol rather than a previous study? Is that correct?

Apologies for the confusion. The usability and acceptability testing was to see if patients and health professionals liked and could use the apps and this is NOT reported in this paper is to be reported elsewhere. The feasibility trial reported in this paper tests whether it is feasible to use as part of practice and whether our trial procedures etc are feasible. The previous study on usability and acceptability is just about to be submitted to a journal but has not yet been published. Publishing the protocol was seen as the priority before too many participants had been recruited. Therefore the ISRCTN number was included as there is more information about the previous study in there. It has been removed to avoid confusion.

Line 24 states that a secondary aim is to assess whether technology based outcome measures are reliable (compared to what?). How will the reliability assessment be undertaken?

The instrumented versions will be compared to the standard methods. So for example the mTUG compared to the standard TUG (professional and stop watch), falls calendars versus falls detector and self-report via app versus EARS questionnaire. This has been made clearer.

Methods

Trial design - now incorporates economic analysis which is not a feature of a feasibility studies? you are not trying to determine cost-effectiveness at this stage and it does state that yet you are collecting HRQoL, healthcare resource use etc. and mentioning economic analyses.

The health economics component was added to the original simple RCT design at the behest of the funding panel. As part of a feasibility study it is important to assess whether it is feasible to collect outcome measures including the health economic ones and to assess the acceptability of the assessments (e.g. number of assessments and time taken). Interestingly, feedback from participants indicates that they do not like the ICE-CAP-O, providing further justification for the inclusion of these health economic questionnaires in the feasibility study. The wording has been amended to ensure it is clear we are not carrying out cost-effectiveness analysis at this stage, merely assessing feasibility of methodology.

Table 1 does not have the trial registration number.

This has been added.

Table 1 does not clearly specify that the study intervention is a smartphone with an application. Surely the trial intervention(s) is provision of a smartphone with goal-setting versus provision of a smartphone without goal-setting?

This has been made clearer. The phone provided to the control group is for outcome measures only, not part of the intervention.

The description of the falls prevention service is irrelevant as it is identical in both arms.

Although this is true the length of service received is a confounding factor and we felt providing this information provided further context. However, we have shortened the description.

Are people actually provided with a smartphone - that wasn't clear to the reader until page 9 where it describes the control arm receive will also receive a Samsung Galaxy with an app? Also healthcare professionals receive a smartphone?

This has now been stated earlier on in the text on page 5 and made clearer in Table 1. The health professionals also receive a phone and use that to goal set and send/receive feedback.

What about contamination here - has that been considered?

The people in the control group are not added to the health professionals' app. So the HCPs cannot goal set with the control group nor send them feedback nor see what exercises they report (only the research team can see that), thereby minimising risk of contamination. The fact that both arms have a smartphone minimises the risk of unblinding with the independent assessors and, we would argue, also reduces the risk of drop-out. There is the potential for the control group to become motivated by reporting their activities. However, there is also the risk of any difference between groups being a function of differential reporting schedules rather than a function of the intervention per se. (see e.g. MERIT study on outcome measurement). If we did not ask the control group to report there was also a risk of reporting fatigue (intervention group asked to report twice via smartphone and questionnaire and control group only completing questionnaire). This has been considered and is an important part of the feasibility trial!

It is not mentioned under the intervention in the flowchart diagram hence difficult for the reader to follow what is being given as the intervention/control. The control states allocated to usual care - but they are given a smartphone with an app thus it is comparing 2 active interventions and there is no "usual care" comparison group?

The smartphone is only being used in the control group as an outcome measure. However, this been made clearer in the diagram as requested.

Under study type it is described as a phase II clinical trial.

This has been amended to state feasibility trial.

Under design, unclear why block randomisation is being used for 30 patients and stratification by age, site when it is feasibility unless the data from this study are feeding into a definitive RCT as internal pilot data.

Stratification is by gender and site to ensure equal distribution across sites e.g. Trafford do not recruit far more men than women and we do not end up with more men in the intervention group. Again, we are testing all trial procedures before a full trial. This was a procedure suggested by the funding panel.

Under page 11, intervention fidelity checks and process evaluation - this is actually an outcome of your feasibility study. Again, written as a definitive RCT in sections which is confusing.

These factors have been moved to outcomes and some of the detail removed this should now be clearer.

Primary outcomes

"characteristics of proposed outcome measures e.g. reliability of falls detector" - how is this being determined?

It will be compared to falls calendars and this has been added.

Under secondary outcomes, it describes that the smartphone will be on a waistband and will act as a falls detector. This is not described under the intervention section - thus it is a specialised device rather than a phone. Unclear how the falls app will be compared to self-report falls data. Will both arms have this facility?

Yes, the control arm has the falls detector and a self-report (outcome only) app. It is not described under the intervention as it is an outcome measure but we have made this clearer in the text.

Seems to be another sub-study as well with another medical device for the TUG - that's fine but these should be clearly stated in the aims and objectives section rather than introduced within the outcomes measurement.

This has been now added as a secondary aim.

Health economics

Why record participant healthcare resource use etc. when the purpose is to work out how much additional costs are required for training in the use of this new technology.

Please see comment above. Health economic data are collected to assess the feasibility of the assessments with patients- are participants willing to complete the data collection forms and how much missing data do we get.

Similarly under the analysis section, why are you reporting an intention to treat approach when this is feasibility work, also adjusting for baseline characteristics - highly unusual for feasibility as you don't have a large enough sample for regression analyses?

We are interested to see whether there is an indication of difference between groups when we run descriptive statistical analyses when using per protocol sample and intention to treat analyses- e.g. does it seem that the smartphone may prove in a definitive trial to be effective for those who actually use it?

However, you are correct and we have removed some of this section to reflect your comments.

I have been fairly critical although am very supportive of the study and the concept, the description of the design and methods could be improved. The team are experts in falls prevention but further clarity would improve the reporting of this study protocol. Good luck with this important study.

Thank you- your comments have been constructive and helpful.

Reviewer: 3

Reviewer Name: Dr Alice Richardson

Institution and Country: Australian National University, Australia Please state any competing interests or state 'None declared': None declared

Please leave your comments for the authors below In this paper, the authors have presented the protocol for a research project to study the feasibility of smartphone technology to prevent falls in older people.

Three items from the Review Checklist (Q9 – Q11) are not applicable in my view because the paper is a protocol not a full study.

Below I will elaborate on the one item in the Review Checklist where my response was “No”. My concerns are of a relatively minor nature and so I recommend that the paper undergo minor revision before resubmission.

Q7: the statistics are dealt with in a somewhat conflicting manner. Firstly, the abstract states that the analyses will be “primarily descriptive” and yet the analysis section (page 15 – 16) mentions a number of inferential methods. The authors do note that the inference will be treated with “great caution” but they could state this before the analysis methods are listed, not after. Secondly, the section on the economic analysis on page 14 seems underdeveloped in comparison to the naming of several instruments to measure quality of life and so on. The authors could be more specific about the tools planned for the economic analysis. Thirdly, I am concerned that the primary outcome will be too rare in the sample size for any meaningful descriptive statistics to be calculated. The authors could give an indication from previous research what the typical fall rate in community-dwelling older persons is to satisfy the reader that a sample size of about 70 will be sufficient for the outcome measures planned.

Thank you very much for your comments. Based on feedback from reviewer 2 we have removed the section about inferential statistics in the analysis section as we agree it is confusing and such statistical analyses are not normally expected from a feasibility study. Because this is feasibility RCT there will not be a full economic evaluation. We have moved some of the analysis from outcome measures to the analysis section to make this clearer. We trust this addresses the third reviewer's comments.

The number of falls in this sample will be too low to detect a significant change in this study and for the purpose of this study we are collecting it only to assess feasibility. 1 in 3 older people fall over the age of 65, but these patients are at high risk of falls with most having fallen before in the last 12 months. Two thirds of people falling will fall again over the following year.

Thankyou for the opportunity to be part of the academic referring system in this way.

Reviewer: 4

Reviewer Name: Edward Meinert

Institution and Country: University of Oxford - United Kingdom Please state any competing interests or state 'None declared': None declared

Please leave your comments for the authors below The authors present a novel study to investigate the feasibility of using a digital health app to support adherence to fall rehabilitation/strengthening programmes. As the authors note, the use of these technologies can serve as a mechanism to provide scalable delivery of monitoring and adherence, leading to better outcomes.

The challenge in this type of research is that the evidence on how to best implement these technologies is limited. Additionally, there is a tendency to focus first on the utility of the technology and not enough emphasis on first considering user needs. The authors have taken care to test and

iterate in the construction of their intervention and now seek to do more rigorous testing on feasibility. Critical review comments herein are centred on providing detail as to the intervention design and subsequent details on the study approach; I am confident the authors have such detail to hand and will be able to enhance the manuscript to reflect their efforts to date and sets a strong foundation for the study.

1. Page 5, rows 25 through 28 - what is delivered by smartphones to justify the use of this technology in this population? What barriers from the literature have the authors identified and what differences could the target population have in their use of this technology which could make adoption challenging? While smartphones could be the solution, is there sufficient evidence to suggest that there is enough saturation of smartphone use in this demographic to justify the cost/development of such technologies for this purpose? What alternates have been considered?

Thank you for your comments, we have added some brief information to the background section in response to your comments (considering word limitations). Adoption within this population is challenging. However, older adults are increasingly using smartphone devices and they are more appropriate than tablets as they can be worn on the person leading to the ability to receive feedback immediately and the ability to monitor falls. Adequate support and something which is simple but tailored to the individual is important for adoption and has emerged from our usability testing and the literature.

2. Page 6, rows 8 through 32 - what was the result of your previous usability and acceptability testing? What did you learn from this work and how did this impact subsequent application development and system implementation?

We have kept this brief as a separate paper on the app development and usability testing is being submitted for publication and also due to word limit. But further information is added on page 7. The majority of changes suggested after usability testing were related to the health professional app and all have been adopted.

3. Overall comment (introduction/background) - it would be useful to have figures illustrating the overall system enterprise architecture and an overview of the system functionality/user interface. Though I do note you speak about the intervention later in the manuscript. It is ok to do this there, but the same comment applies as the details provided in the Intervention section do not provide detail to what has been developed, nor the way the system has improved over time via feedback to inform the current state of the study.

This is difficult to add as is included and described in a separate paper, some detail has been added about improvements (see above). We also include figures under supplementary material.

4. Page 7, row 42 (interventions) - what impact does the variant durations have on results? What is the length of the intervention use including the digital health app intervention?
The impact of delivery time should make no difference on results as it is the same across intervention and control. The app is used for the full 6 months, however, participants in Trafford will not receive personalised messages after discharge at 8 weeks only automated messages (they are asked to goal-set for patients until the end of their follow-up period in the study at discharge). Differences between sites will be explored and this is mentioned on page 13.

5. Page 8, row 9 (Inclusion and exclusion criteria) - how has sampling been completed to ensure demographic saturation (e.g. education, socio-economic, ethnicity, etc.)

As this is a pragmatic trial we wish to recruit all patients eligible who would go through a service. The two sites have a mix of demographics and provide a good range socio-demographic spread, and we have added a brief reference to this on page 8. From recruitment so far we have a good representation of education, socio-economic background and ethnicity!

6. Page 8, rows 48 through 59 (eligibility) - how have the authors taken into account the difference in data performance from 3G to 4G to Wifi. How will this be assessed and what criteria used to define "good"?

We document whether they have wifi or 3/4G in the CRF and health professionals take a study phone with them to the patients' house when first discussing the study or the researcher takes a study phone with them before taking consent. Connectivity is 'good' and sufficient for the app to work if you can access webpages whilst in the person's home. This has been made clearer in this section.

7. Page 9, row 13 and 14 - what is the reference for the point regarding peer work and it is unclear how peer work will be used in the study or is a variable which is monitored.

We have clarified this point, we do not have evidence to prove this point but think it has the potential to assist recruitment. It has also been added on page 13 as an outcome.

8. Page 10, row 22 - why was this particular smartphone selected for the study?

This has previously been used in our usability testing and also a previous European study and shown to have good usability and adequate specification for the falls detector. This was stated in the original manuscript but removed because of word count, we have added it back in.

9. Page 10, row 28 - who developed the motivate me app and how is continuous application development and support for the technology administered and planned throughout the study?

Both apps were developed by mHealth Technologies srl, a University of Bologna spin off company, and any technical support and app updates are provided by them. However, technical support with patients is provided by the lead author in consultation with mHealth Technologies. The lead author has made several visits to Bologna as part of the development process. Details have been added on page 9.

10. Page 12 (outcome measures) - what will happen if you do not receive demographic saturation of participant types?

The main aim is to ascertain whether it is feasible for patients attending falls rehabilitation. If recruitment of patients is not representative of patients attending rehabilitation then this will be an important finding related to feasibility.

11. Page 14, row 33 - how will you ensure the participants use the smartphones as you envisaged? Is it realistic to think that participants will always use a smartphone in this manner; do you have previous evidence from your previous work that indicates the likelihood of the success of this?

We have evidence from the usability testing that people are willing to report their exercises. Health professionals were more sceptical so we tested it with patients and also carried out qualitative work with them. However, this is part of the feasibility of using a self-report app as an outcome measure - see outcomes (page 13) and analysis section (page 17).

12. Page 17, row 57 - you are offering participants to use a new mobile phone that is not their pre-existing mobile phone; how have you accounted for the difference in the functionality of potentially more sophisticated devices than their current devices?

Participants have shown they are able to use the app and smartphones even if they have never used one before, the intervention app will run on all android phones. We had to provide phones because they needed to be a certain specification for the falls detector and also due to confidentiality reasons for the trial. We have added the fact that they are not using it on their own phone as a potential limitation page 19.

13. It would be useful to see additional references to recent smartphone research which will have extensibility to this study both in study design and intervention design.

Additional references have been included in the introduction where possible and within the word limit.

VERSION 2 – REVIEW

REVIEWER	Dr Alice Richardson Australian National University, Australia
REVIEW RETURNED	27-Jun-2019

GENERAL COMMENTS	In the revised version of this paper, the authors have addressed a number of issues throughout the paper. Three items from the Review Checklist (Q9 – Q11) are not applicable in my view because the paper is a protocol not a full study. I am satisfied that the authors have addressed the concerns regarding conflicting statistical analyses, and so I am recommending that the paper be accepted. I look forward to reading about the results of the trial. Thankyou for the opportunity to be part of the academic referring system in this way.
--

VERSION 2 – AUTHOR RESPONSE

Thank you for these final additional comments. We have added in further detail about the risk of contamination on pages 11 and 19, and around stratification by gender and site on page 8. We have also completed the SPIRIT checklist.

As this is an NIHR funded study we need to inform them of publication 28 days beforehand, we trust that this is acceptable.